# Nested non-covalent interactions expand the functions of supramolecular polymer networks

David J. Lundberg[1], Christopher M. Brown [2], Eduard O. Bobylev[2], Nathan J. Oldenhuis [3], Yasmeen S. Alfaraj [2], Julia Zhao[2], Ilia Kevlishvili [1], Heather J. Kulik [1] & Jeremiah A. Johnson [2,4] ✉

Supramolecular polymer networks contain non-covalent cross-links that enable access to broadly tunable mechanical properties and stimuli-responsive behaviors; the incorporation of multiple unique non-covalent cross-links within such materials further expands their mechanical responses and functionality. To date, however, the design of such materials has been accomplished through discrete combinations of distinct interaction types in series, limiting materials design logic. Here we introduce the concept of leveraging "nested" supramolecular crosslinks, wherein two distinct types of non-covalent interactions exist in parallel, to control bulk material functions. To demonstrate this concept, we use polymer-linked $Pd_2L_4$ metal–organic cage (polyMOC) gels that form hollow metal–organic cage junctions through metal–ligand coordination and can exhibit well-defined host-guest binding within their cavity. In these "nested" supramolecular network junctions, the thermodynamics of host-guest interactions within the junctions affect the metal–ligand interactions that form those junctions, ultimately translating to substantial guest-dependent changes in bulk material properties that could not be achieved in traditional supramolecular networks with multiple inter-actions in series.

Supramolecular polymer networks are formed through non-covalent bonds (e.g., metal–ligand coordination, hydrogen bonding, host-guest binding) between (macro)molecular strands and junctions, leading to a wide range of network structures and properties from elastic mate-rials to highly ordered framework materials. Compared to conven-tional, covalently cross-linked polymer networks, the reversibility of the non-covalent interactions used to construct supramolecular polymer networks enables distinct functionalities (e.g., self-healing[1], stress dissipation[2], responsiveness to chemical stimuli[3]) and enhanced mechanical properties (e.g., extensibility and toughness[4,5]), which

facilitate a variety of applications, such as adhesives[6], biomaterials[7], and stimuli-responsive materials[3]. In some cases, non-covalent bonds form discrete supramolecular structures (e.g., metallacycles or metal–organic cages from metal–ligand coordination bonds or extended pi-pi stacking clusters), which themselves can serve as cross-links. While such structures can introduce advanced functionality, the mechanical properties and stimuli-responsiveness of the resulting materials are generally defined by the underlying non-covalent bonds.

Researchers have combined multiple types or strengths of non-covalent bonds to construct new materials with advanced functions

[1]Department of Chemical Engineering, Massachusetts Institute of Technology, 77 Massachusetts Avenue, Cambridge, MA, USA. [2]Department of Chemistry, Massachusetts Institute of Technology, 77 Massachusetts Avenue, Cambridge, MA, USA. [3]Department of Chemistry, University of New Hampshire, 23 Academic Way, Durham, NH, USA. [4]David H. Koch Institute for Integrative Cancer Research, Massachusetts Institute of Technology, 77 Massachusetts Avenue, Cambridge, Massachusetts, USA. ✉e-mail: jaj2109@mit.edu

(Fig. 1A). In these examples, however, the interactions used (*vide infra*) have been introduced as discrete, isolated interactions where the (thermo)dynamics of one are independent of the other within the material structure[8–12]. For example, Stang and coworkers have used combinations of orthogonal metal–ligand coordination (e.g., Pt[2+]-pyridine metallacycles), host–guest (e.g., ammonium/crown-ether complexes), and hydrogen-bonding interactions to construct multi-responsive supramolecular polymer networks[13–16] that display self-healing behavior and stimuli-responsive gel-to-sol transitions under conditions that weaken relative non-covalent bond strengths or increase dynamics (e.g., heating) or dilute cross-linking functionality (e.g., the addition of potassium salts to sequester host-guest cross-linking sites). Since host-guest binding, hydrogen bonding, and metal–ligand coordination interactions are independent in such systems, attenuating any of the interaction types leads to network deconstruction. Moreover, strengthening only one non-covalent interaction type does not increase the overall network stability to a competitor for the others[17]. Similarly, others have reported materials that display pH or solvent responsiveness made possible by a combination of host-guest and electrostatic or hydrogen-bonding

interactions[9,18]. Nevertheless, while materials built from host-guest interactions intrinsically have the potential for highly selective stimuli-responsive bulk behaviors triggered by the presence of selective guests, such stimuli have so far been strictly limited to causing cross-link displacement resulting in a decrease in stiffness or network dissolution[19,20]. We note that while host-guest binding can occur through ensemble non-covalent interactions (e.g., hydrogen bonding, hydrophobic interactions, and van der Waals forces), these interactions are defined by the host/guest pair and are not independently modifiable.

Others have combined non-covalent interactions of the same type but varied strength (e.g., a weak and a strong coordination bond) to fabricate materials (Fig. 1A, right). For example, Holten-Andersen and coworkers, as well as Craig and coworkers, have prepared end-linked or side-chain linked metal–ligand coordination gels, respectively, by combining mixtures of metal ions (e.g., Ni[2+], Zn[2+], Cu[2+], Pd[2+], Pt[2+]) with a polymer-bound ligand[21–24]. A similar concept has recently been demonstrated in the context of Pd[2+] and Pt[2+] metal-organic cage (MOC)-crosslinked polymer networks[25]. Nevertheless, while the combination of multiple supramolecular bond strengths through mixing

## A Previous Work: Discrete Combinations of Multiple Non-Covalent Interactions

Host–Guest, Metal–Ligand Coordination,
and Hydrogen Bonding Interactions (Stang)

Multiple Strengths of an Interaction Class
e.g., Metal–Ligand Coordination (Holten-Andersen, Craig)

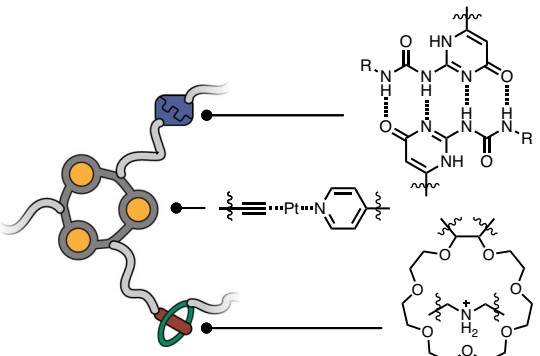

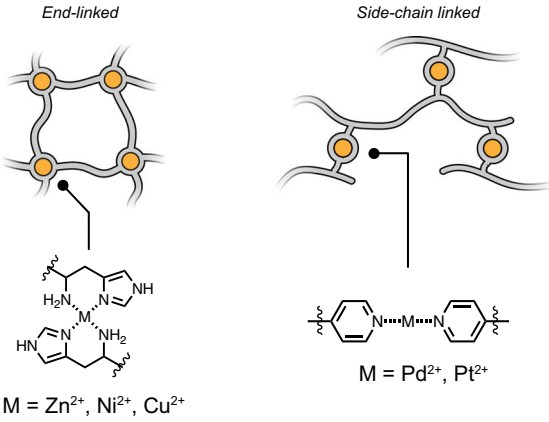

+ Broadened functionality and stimuli-responsiveness
- Traditional stoichiometric assembly limitations
- Supramolecular interactions are independent

+ Tunable dynamic properties
- Traditional stoichiometric assembly limitations
- Maintained stimuli-responsive properties

## B This Work: Nested Metal–Ligand Coordination and Host–Guest Binding

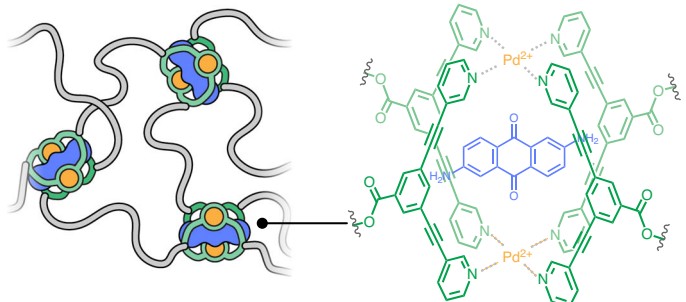

Nested Configuration Enables:

+ Modulation of global dynamics with guest strength

+ Expanded stoichiometric assembly window

+ Small-molecule triggered sol–gel transitions

+ Guest binding strength-selective sol–gel transitions

+ Self-sorting driven sol–gel transition networks

**Fig. 1 | Examples of supramolecular networks constructed with multiple interaction types or strengths. A** Previous examples of discrete combinations of multiple supramolecular interactions within polymer networks. Multiple distinct interaction types allow expanded functionality and stimuli-responsiveness but ultimately suffer from the traditional limitations of supramolecular interactions in general (e.g., stoichiometric assembly limitations). Additionally, combinations of multiple strengths of the same interaction class enable exquisitely tunable dynamic properties but introduce no further network stability or stimuli-responsiveness. **B** PolyMOC gels reported here leverage the nested nature of metal-ligand coordination and host-guest interactions within their junctions to enable enhanced and expanded mechanical and stimuli-responsive properties.

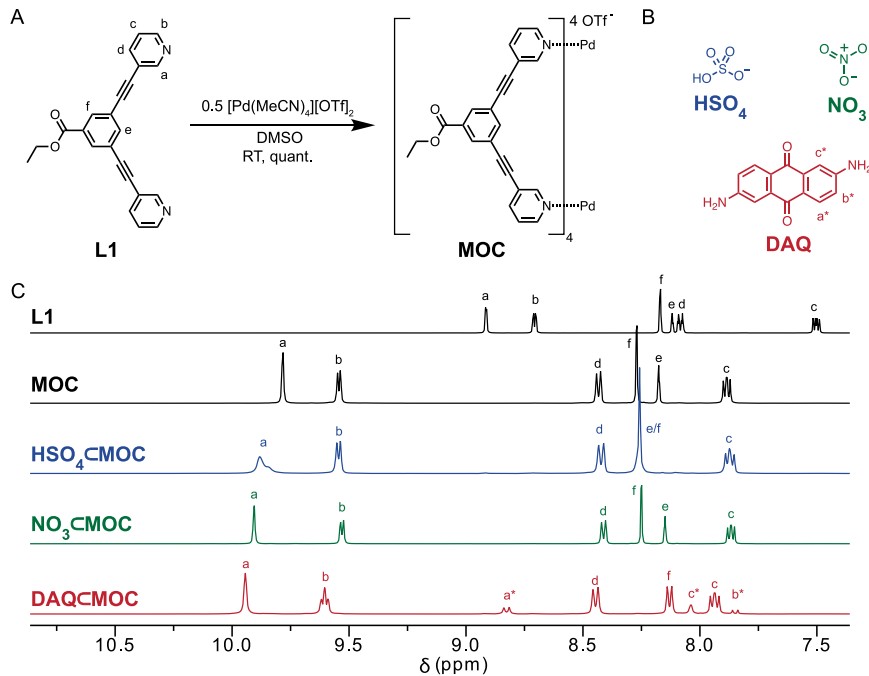

**Fig. 2 | Synthesis and characterization of MOC and corresponding host-guest complexes. A** Synthesis of MOC-OTf from L1 and [Pd(MeCN)₄][OTf]₂ in DMSO. **B** Chemical structures of the three guests used in this study: tetrabutylammonium salts of hydrogen sulfate (**HSO₄**), nitrate (**NO₃**), and 2,6-diaminoanthraquinone (**DAQ**). **C** Partial $^1$H NMR spectra of **L1, MOC-OTf, HSO₄⊂MOC, NO₃⊂MOC**, and **DAQ⊂MOC** for 1 equiv. of guests added per cage (full spectra are shown in Supplementary Fig. 25–28). Distinct down-field peak shifts relative to **L1** indicate successful MOC formation and subsequent guest-binding within the MOC cavity.

metals enables functions such as constant stiffness with widely variable stress relaxation rates, the reliance on independent interactions limits the functions of these materials similarly to those using different supramolecular interactions as described above. Overall, the use of combinations of independent non-covalent interactions in series places inherent limits on the functional scope of supramolecular polymer networks.

We envisioned a new approach to controlling bulk supramolecular material properties by leveraging multiple supramolecular interactions in a nested or parallel configuration, such that the two interactions are not independent, as in the examples described above, but dependent on each other (Fig. 1B). In searching for a material to demonstrate this concept, we recognized that such a configuration could be achieved with MOC-crosslinked polymers (polyMOCs)[26–29]. MOCs are, in some cases, known to display pronounced host-guest binding ability[30–34], and a few reports of polyMOCs have leveraged MOCs that can bind guest molecules[35–38]; however, no studies have explored how the bulk properties of polyMOCs can be varied by altering guest binding thermodynamics, nor have they leveraged this unique nested junction structure to achieve functions that cannot be obtained with traditional supramolecular networks. For example, we imagined at least three functions that could potentially be uniquely achieved with nested polyMOC networks: (1) guest-induced manipulation of bulk stress-relaxation dynamics, wherein small molecule guests alter the metal–ligand coordination dynamics of the junction; (2) resistance to dissolution under off-stoichiometry network formation conditions or in the presence of competitive reagents due to synergistic stabilization of MOC junctions bound to guests; and (3) the ability to control both percolation (i.e., gelation) and dissolution using selective self-sorting of MOC junctions driven by guest recognition. Here, we realize the synthesis of a new class of polyMOCs containing acetylene-spaced Pd₂L₄ junctions and demonstrate each of the above functions using both neutral and ionic guest molecules, establishing nested junctions as a new design principle for supramolecular polymer networks.

## Results & discussion

Initially, we targeted a Pd₂L₄ MOC based on bis-pyridyl ligands with acetylene spacers due to its extensively studied host-guest binding properties and our previous successful translation of related bis-pyridine MOCs to polyMOC materials. When the bis-*meta*-pyridine ligand **L1** is mixed with 0.5 equiv [Pd(MeCN)₄][OTf]₂ in DMSO (5 mM Pd$^{2+}$) at room temperature, supramolecular complex **MOC** is formed rapidly and quantitatively (Fig. 2A). Although no guest molecule is explicitly added, the triflate counterion is presumed to reside within the MOC cavity (*vide infra*). Here, we note that coordinating solvents (e.g., DMSO or acetonitrile) are necessary for MOC formation and subsequent polyMOC fabrication based on this MOC. MOCs of this structure are known to strongly bind guests with electron-rich substituents within their cavity, including (divalent) anions[39] or quinone-type molecules; however, previous studies have largely focused on quantifying association constants within structurally similar Pd₂L₄ MOCs bearing non-coordinating counterions such as tetrakis[3,5-bis(trifluoromethyl)phenyl]borate (BArF), cages in non-polar solvents (e.g., dichloromethane, nitromethane), or a combination thereof[40]. Therefore, direct translation of previously reported extended-quinone guest molecules proved unsuccessful, as either their binding strength with **MOC** is too low (in the case of naphthoquinone, $K_a \sim 25\,M^{-1}$), or the guests exhibit insufficient solubility in DMSO (as was the case with anthraquinone and pentacenequinone). Thus, we identified three previously unreported guests for **MOC**: tetrabutyl ammonium salts of hydrogen sulfate, **HSO₄**, or nitrate, **NO₃**, and 2,6-diaminoanthraquinone, **DAQ** (Fig. 2B). When 1 equiv of each guest is added to a 5 mM DMSO-$d_6$ solution of **MOC**, distinct down-field peak shifts are observed via $^1$H NMR spectroscopy (Fig. 2C). In line with previous reports, the interior proton resonances of **MOC** ($H_a$ and $H_b$ based on labeling in Fig. 2A) show the largest shifts upon guest addition, indicative of binding *within* the MOC cavity. Association constants, $K_a$, for guest binding were quantified through titration experiments and are listed in Table 1 (see Supplementary Fig. 1–6). The $K_a$ measurements fit well to a 1:1 binding isotherm and span approximately three orders of

## Table 1 | Measured Association Constants for guests with MOC

| Guest | Association Constant (M⁻¹) |
|---|---|
| HSO₄ | 1000 ± 500 |
| NO₃ | 6000 ± 1000 |
| DAQ | 62,000 ± 2000 |

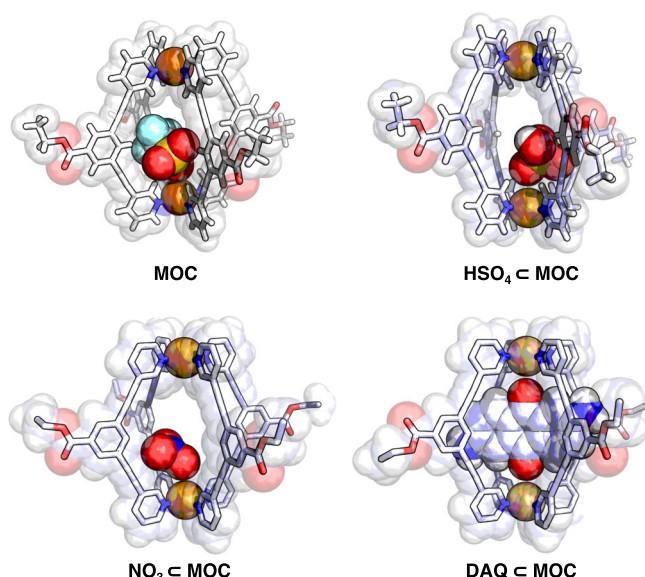

**Fig. 3 | X-ray crystal structures of MOCs illustrating the presence of guest within the cavity: MOC, HSO₄⊂MOC, NO₃⊂MOC, and DAQ⊂MOC.** Counter ions and solvent molecules have been removed for clarity.

magnitude, with binding strength increasing in the order: **HSO₄** < **NO₃** < **DAQ**. To our knowledge, the association constant of 62,000 ± 2000 M⁻¹ for **DAQ** is the largest reported for any Pd₂L₄ MOC in DMSO by three orders of magnitude. While we could not quantify the binding strength of the triflate anion within the cavity of this cage due to its inherent presence in the palladium salt used, we performed density functional theory calculations of its binding strength using the B3LYP-D3(BJ) method[41,42], which predict that the triflate anion will bind within a guest-free **MOC** with a favorable free energy difference of 128.4 kJ/mol, suggesting near quantitative guest occupancy (triflate or otherwise) under the conditions explored in this work.

To further probe the nature of these guest binding interactions within **MOC**, single-crystal X-ray diffraction studies of **MOC** and its three host-guest complexes were performed. The crystal structures of **MOC**, **HSO₄⊂MOC**, **NO₃⊂MOC**, and **DAQ⊂MOC** are shown in Fig. 3. In all cases, the added guest molecules reside within the cavity of the MOC, despite the variable ionic nature of the guests explored. Consistent with ¹H NMR studies, hydrogen bonding interactions are observed between guest molecules and interior proton H$_a$, with bond distances ranging from 2.35 to 2.75 Å for all added guests, suggesting that binding of these guests is not exclusively driven by partial ion exchange. Interestingly, guest orientation was consistent with that of MOC packing for both **OTf⊂MOC** and **DAQ⊂MOC** (i.e., guest molecules were oriented in a consistent fashion throughout the crystal), perhaps due to their size relative to the MOC cavity; however, the position of **NO₃** and **HSO₄** within their respective hosts was observed to be split between 'top' and 'bottom' sites, as judged by equal electron density in both positions, but with relative intensity ~half that of the MOC structure, consistent with 1:1 binding as observed from titration experiments. Despite having the highest measured K$_a$, **DAQ** shows the longest hydrogen bonding distances; complementary ArH···π

interactions are observed between H$_e$ of the MOC and the flanking aryl rings of **DAQ**. This ArH···π interaction is hypothesized to drive the stronger binding and larger ligand distortion and desymmetrization of the cavity in **DAQ⊂MOC**, compared to **MOC**, **HSO₄⊂MOC** and **NO₃⊂MOC**, as judged by the angles between pyridine ligands (an average deviation from 90° of 1.86° as compared to 1.03°, 1.07°, and 0.36°, respectively. See Figures S36-40 for more details). Such desymmetrization illustrates the flexural ability of MOCs to change their structure to selectively bind guests[43].

Having characterized the binding of **HSO₄**, **NO₃**, and **DAQ**, we next sought to verify that the same host-guest interactions exist in an analogous polyMOC gel. A series of four polyMOC gels were fabricated from polymer ligand **PL1**—which contains **L1** ligands appended to the ends of a linear poly(ethylene glycol) chain (M$_n$ = 4600 Da)—in DMSO (5 wt.% **PL1**), with or without the inclusion of guest (Fig. 4A). Gels were annealed for 2 h at 60 °C with inclusion of sufficient equivalents of each guest to achieve >95% binding occupancy (see Supplementary Table S1). This procedure yielded robust, free-standing materials (Fig. 4A) in all cases, which we refer to as **poly(XMOC)**, where X indicates the molecule residing within the MOC junctions: **OTf, HSO₄, NO₃, DAQ**. Cross-polarization magic-angle spinning (CP-MAS) ¹H NMR spectroscopy of the polyMOC gels shows complete MOC assembly in all cases and distinct down-field resonance shifts when guests are added, as compared to the parent **poly(OTfMOC)** (Fig. 4B). The relative peak shifts from CP-MAS ¹H NMR are in-line with those observed in solution for discrete host-guest complexes, suggesting that the presence and nature of guest binding remain similar within the polyMOC gels. Moreover, resonances associated with **DAQ** are present in the **poly(DAQ⊂MOC)** spectrum, further supporting the formation of a host-guest complex, effectively immobilizing **DAQ** within the MOC junctions of the material. The mechanical properties of each polyMOC gel were probed through shear rheology frequency sweep experiments. The gels exhibited near identical storage moduli (G′) of 7 kPa at 1 rad/s (Fig. 4C), indicating that the presence or absence of added guest molecules does not noticeably change junction assembly or network topology[44].

Additionally, we studied the ability of pre-formed polyMOC gels to bind guests from solution. When a 1 mL puck of **polyMOC-OTf** was placed into a solution of **DAQ** (10 mM in DMSO-d⁶), within 60 s, the surface of the gel began to darken, suggesting diffusion of **DAQ** into the gel. After 3 h, the gel was characterized by CP-MAS ¹H NMR; the spectrum displayed similar chemical shifts to as-assembled **poly(DAQ⊂MOC)** (Supplementary Fig. 36), confirming **DAQ** diffused into and formed a host-guest complex within the junctions of the network. This result suggests that guest binding does not require the elevated temperature and increased ligand exchange dynamics present during the fabrication of these polyMOC gels.

Next, we sought to determine if the inclusion of guest-binding within the junctions of polyMOC gels would affect their stress relaxation dynamics. Stress relaxation dictates the temporal response of a material to applied forces and is a key parameter in the design of polymer networks for use as adhesives, injectable biomaterials, and cell scaffolds[7]. We reasoned that the inclusion of host-guest binding may serve to stabilize the MOC junctions against the solvent- or ligand-assisted exchange reactions at palladium that lead to microscopic network restructuring and, ultimately macroscopic stress relaxation[45]. Stress-relaxation studies conducted at 25 °C show that the presence of host-guest binding causes an increased characteristic relaxation time concomitant with the strength of guest binding, up to approximately an order of magnitude greater for **poly(DAQ⊂MOC)** as compared to **poly(OTf⊂MOC)** (Fig. 5A). To gain further insight into how guest-binding affects stress relaxation, variable temperature experiments were performed, and an Arrhenius analysis was conducted (Fig. 5B). The extracted activation energies for each gel are listed in Table 2. Activation energy was found to trend upward with guest binding

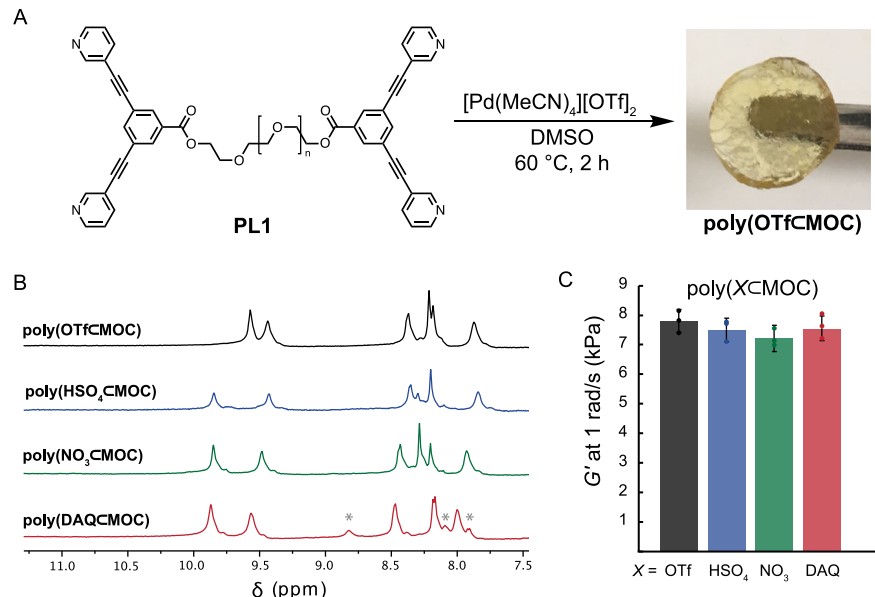

**Fig. 4 | Fabrication and characterization of $Pd_2L_4$ polyMOC gels. A.** A polymer ligand (**PL1**) is combined with $[Pd(MeCN)_4][OTf]_2$ in DMSO. Annealing provides **poly(OTf⊂MOC)**, which is composed of $Pd_2L_4$ **MOC** junctions. Additional guest molecules can be included before annealing to yield similar results. **B** CP-MAS $^1H$ NMR spectra of polyMOC gels fabricated with and without added guests. Relative resonance shifts of $H_a$ within polyMOC gels are in line with those of the analogous MOCs (see Fig. 1A), suggesting the guest binding remains unchanged when translated to a polyMOC gel. Resonances marked with an asterisk in the **poly(DAQ⊂MOC)** spectrum belong to cage-bound **DAQ**. Full spectra are shown in Figures S25–S28. Slight shoulder peaks may be due to MOC clustering within the gel that could modify guest-binding behavior. **C** Comparison of gel shear modulus (G') as measured at 1 rad/s. Inclusion of guest has no statistically significant impact on modulus. Full frequency sweep data are shown in Supplementary Fig. 29. Error bars represent ±1 standard deviation.

strength, indicating that the critical ligand dissociation step required for stress relaxation has a higher activation barrier for more stable host-guest complexes. This result highlights the ability of host-guest binding to stabilize the metal-ligand coordination interaction through the unique nested configuration present in this system. Such behavior contrasts with the properties observed for previously reported supramolecular gels with multiple discrete interactions, where, upon the incorporation of additional interaction types and strengths, the terminal relaxation mode is unaffected[24]. Further studies regarding these effects are ongoing in our laboratory.

Having confirmed guest binding within the junctions of polyMOC gels and its ability to modulate the rates of ligand-exchange reactions at the molecular level, which translate to bulk stress relaxation differences, we hypothesized that guest inclusion could provide a means to stabilize polyMOC gels under conditions that are typically incompatible with network formation, such as in the presence of super-stoichiometric metal ions. Stoichiometry is critical for the formation of both covalent and supramolecular polymer networks, and previous studies have explored the impact of off-stoichiometry components on metallogel properties[46,47]. With sub-stoichiometric metal, cross-linking density is proportional to metal ion concentration; when there is excess metal, incomplete, lower-functionality metal-ligand complexes form, precluding gelation. In our polyMOCs, addition of super-stoichiometric $Pd^{2+}$ would therefore be expected to drive MOC disassembly and produce free ligands coordinated to two palladium ions (Fig. 6A). To test the potential ability of guests to stabilize MOCs under these conditions, assembly of **MOC** was attempted with 4 equiv. of $[Pd(MeCN)_4][OTf]_2$. When using this super-stochiometric $Pd^{2+}$, a mixture of products is obtained as supported by $^1H$ NMR (Fig. 6B, orange, middle spectrum). The $^1H$ NMR spectrum shows a combination of **MOC** (65%) and what we attribute to an **L1·Pd₂** coordination structure (~35%); however, in the presence of 1 equiv **DAQ**, the $^1H$ NMR spectrum of the mixture shows an increased intensity of the peaks attributed to **MOC** and a concomitant decrease in intensity of **L1-Pd₂** (Fig. 6B, red spectrum), indicative of shifting the equilibrium toward MOC

formation (88%). Within the context of a polyMOC gel, this guest-driven stabilization in the presence excess $Pd^{2+}$ could be expected to drive sol-gel transitions if the initial degree of disassembly pushed the system below the gel point, as illustrated in Fig. 6C. Correspondingly, the attempted fabrication of **poly(OTf⊂MOC)** with 4 equiv $[Pd(MeCN)_4][OTf]_2$ resulted in a free-flowing liquid. Addition of 1 equiv **DAQ** to this liquid yielded a robust, free-standing gel with a storage modulus of 2300 Pa. This gel could then be dissolved through the subsequent addition of 3 equiv $Pd^{2+}$. CP-MAS $^1H$ NMR spectroscopy of these materials confirmed guest-binding (see Supplementary Fig. 31–33).

Expanding on this process, three sequential sol-gel transitions were achieved by alternating addition of **DAQ** and $Pd^{2+}$ (Fig. 6D). Both **HSO₄** and **NO₃** can induce a sol-gel transition in polyMOC gels in the presence of excess $Pd^{2+}$, but in line with their lower binding affinities, the extent was less pronounced than for **DAQ**. A sol-gel phase diagram for assembly of polyMOC gels containing 1 equiv of each guest and between 1 and 5 equiv $Pd^{2+}$ was constructed (Fig. 6E), showing that the amount of excess $Pd^{2+}$ that can be tolerated before gelation is precluded is proportional to guest binding strength. Materials with 4 equiv $Pd^{2+}$ underwent a sol-gel transition when 1 equiv of all guests were added, representing an unselective guest-sensing response. In contrast, systems with 4.5 equiv $Pd^{2+}$ only underwent a sol-gel transition in the presence of **DAQ**. Addition of >10 equiv **HSO₄** or **NO₃** under these conditions failed to elicit a sol-gel transition, illustrating the ability to engineer selective guest-triggered sol-gel transitions through varying guest binding affinity.

Motivated by the ability to drive sol-gel transitions upon guest introduction for systems with super-stoichiometric $Pd^{2+}$, we sought to explore a complementary mechanism to expand the scope of selective and predictable material property changes in polyMOC systems (for example, the possibility for both sol-to-gel and gel-to-sol transitions driven by binding of the same guest, as well as reversible phase transitions when selective guest binding is attenuated). Previous reports have demonstrated that guest binding can drive self-sorting within

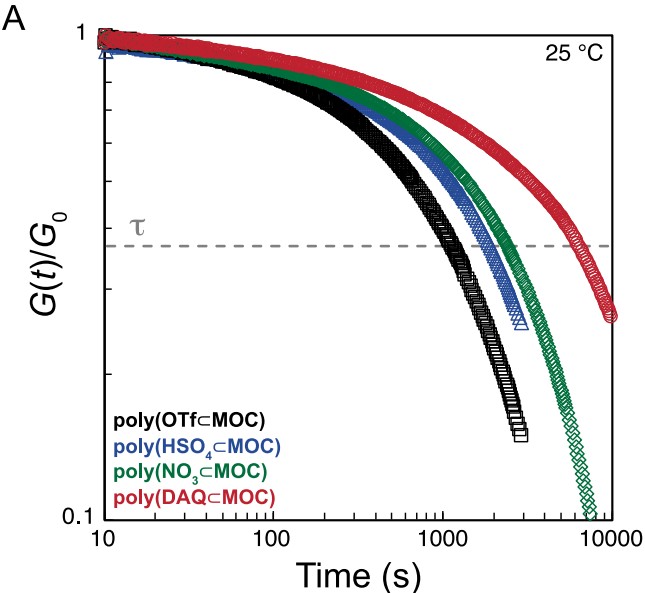

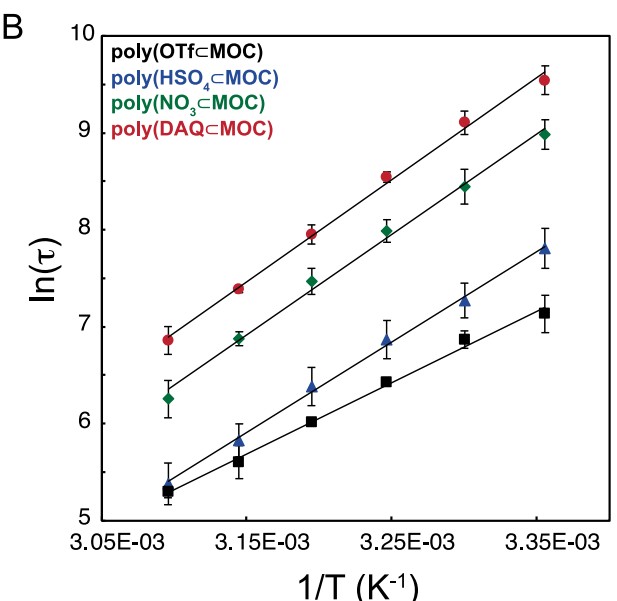

**Fig. 5 | Stress relaxation studies on polyMOC gels. A**. Room temperature step-strain (2%) relaxation curves of **poly(OTf⊂MOC)**, **poly(HSO₄⊂MOC)**, **poly(NO₃⊂MOC)**, and **poly(DAQ⊂MOC)**. **B** Arrhenius plot of stress relaxation time (τ) versus inverse temperature for **poly(OTf⊂MOC)**, **poly(HSO₄⊂MOC)**, **poly(NO₃⊂MOC)**, and **poly(DAQ⊂MOC)**. Thermal activation energies are listed in Table 2. Error bars represent the 99% confidence interval.

heteroleptic MOCs formed from mixtures of chemically distinct but isostructural ligands[48–50]. Specifically, in the absence of guest, statistical mixtures of heteroleptic cages arise; however, when a guest is added, the equilibrium is shifted toward the cage structure that binds the guest most strongly, typically resulting in narcissistic self-sorting (i.e., homoleptic MOCs are preferentially formed). A schematic for such behavior is shown in Fig. 7A for a mixture of **L1** and an isostructural endo-pyridyl variant, **L2**, that on its own will form cages with no binding affinity for **DAQ** (*vide infra*). We reasoned that the same mechanism could be leveraged to produce topological changes in polyMOC gels fabricated with mixtures of distinct polymer-bound and small-molecule ligands that, on their own, would form MOCs of the same architecture, but with varying abilities to bind a specific guest as illustrated in Fig. 7B. If such self-sorting were sufficiently selective, sol-

### Table 2 | Measured Activation Energies for Stress Relaxation in polyMOC gels

| polyMOC Gel | $E_a$ (kJ.mol$^{-1}$) |
| --- | --- |
| poly(OTf⊂MOC) | 61 ± 3 |
| poly(HSO₄⊂MOC) | 78 ± 5 |
| poly(NO₃⊂MOC) | 86 ± 3 |
| poly(DAQ⊂MOC) | 88 ± 4 |

gel transitions could be achieved through the preferential incorporation of polymer-bound ligands into MOC structures *via* a favorable guest binding event (Fig. 7B panel 1, 2). Further, such sol-gel transitions could have the potential to be reversible upon the addition of a guest that non-selectively binds cage structures from the ensemble (Fig. 7B panel 2, 3). With the further addition of the selectively binding guest, another sol-gel transition may be achieved (Fig. 7B panel 3, 4); this cycle could potentially be repeatable if stronger binding or greater equivalents of each type of guest (selective versus non-selective binding) were added.

Guided by previous reports that the endo-pyridyl variant of **MOC** displays attenuated binding of quinone-type guests[51], we synthesized the corresponding endo-pyridyl ligand **L2** (Fig. 7A) to explore its impact on MOC and polyMOC assembly in the presence of added guests and potential to enable self-sorting. Using only **L2** under standard MOC assembly conditions, **N-MOC** was formed quantitatively. Titration experiments showed that, unlike **MOC, N-MOC** displayed no affinity for **DAQ** ($K_a \ll 1\,M^{-1}$) and had a reduced affinity for both **HSO₄** and **NO₃** (association constants of $33 \pm 10\,M^{-1}$ and $2200 \pm 400\,M^{-1}$). This difference in affinity for **DAQ** between **MOC** and **N-MOC** is sufficiently strong to drive considerable self-sorting to preferentially incorporate **L1** into MOC structures when equimolar amounts of **L1** and **L2** (0.5 equiv total) were mixed super-stoichiometrically with 0.375 equiv [Pd(MeCN)₄][OTf]₂ in DMSO (5 mM solution of Pd$^{2+}$). For this system, we first quantified the preferential incorporation of **L1** into MOC structures upon guest addition using $^{1}$H NMR spectroscopy, as $H_a$ and $H_b$ resonances for unbound **L1** and **L2** could easily be resolved. All guests were observed to preferentially drive **L1** incorporation into MOC structures with an extent proportional to the differential guest binding strength between **MOC** and **N-MOC** (Fig. 7C). The greatest shift in ligand incorporation occurred in the presence of >1 equiv **DAQ**, resulting in a 12% increase in the fraction of **L1** in MOC structures, which corresponds to a 50% increase in the selectivity for **L1** incorporation over **L2**. Interestingly, when 3 equiv **NO₃** were added to the mixed ligand system containing 1 equiv. **DAQ**, the ratio of free to cage bound **L2:L1** was quickly shifted (within 15 min at room temperature) from 2:1 to 1.5:1 (Fig. 6C), illustrating the ability to reversibly modulate ligand self-sorting in situ through the introduction of less-selective guests.

Looking to translate this self-sorting effect into a polyMOC gel system, an optimized ratio of polymer-bound and small-molecule ligands was required. Using Macosko-Miller theory[52,53], we modeled the predicted network structures that could be formed through arbitrary combinations of **PL1, L2**, and **Pd$^{2+}$**, and derived gelation criteria based on the proportions of **L2: PL1** and of **L2 + PL1: Pd$^{2+}$**, and a self-sorting preference factor, denoted as $\rho_{L2:PL1}$, $\rho_{L:Pd^{2+}}$, and $\phi$, respectively. Here, $\phi$ describes the preference for incorporation of polymer-bound bis-pyridine ligands into MOC structures and is equal to the fraction of **L1** moieties in MOC structures divided by the fraction of all ligand moieties in the mixture that are **L1**. We derived the following inequality, which predicts gelation to occur when:

$$\left(\frac{1}{\rho_{L2:PL1}+1}\right)\frac{\phi}{2\rho_{L2:PL1}} > \frac{1}{\sqrt{\frac{1+\rho_{L2:PL1}}{\frac{1}{2}\rho_{L:Pd^{2+}}}(f_w-1)}} \tag{1}$$

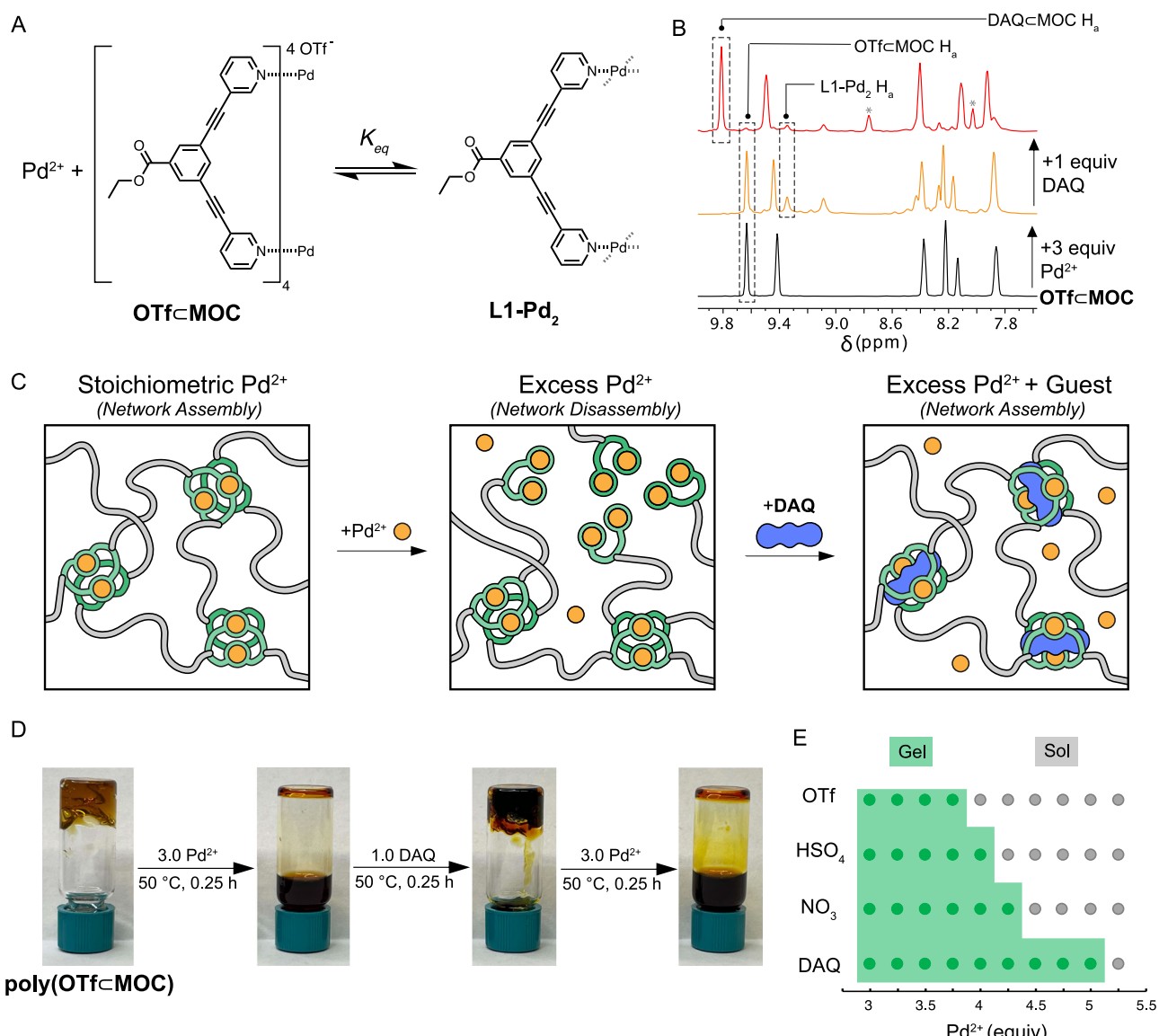

**Fig. 6 | Effect of excess Pd²⁺ on (poly)MOC assembly. A** Schematic of the MOC-ligand equilibrium in the presence of excess Pd²⁺. The addition of super-stoichiometric Pd²⁺ drives the equilibrium to the right, producing free ligands coordinated to Pd (**L1·Pd**). **B** ¹H NMR spectra of **MOC-OTf** assembled on stoichiometry (1 equiv Pd²⁺), and with super-stoichiometric Pd²⁺ (4 equiv) with and without added **DAQ**. **C** Schematic of network degradation in the presence of super-stoichiometric Pd²⁺ and subsequent gelation upon addition of **DAQ**. Super-stoichiometric Pd²⁺ drives MOC structures into L-Pd₂ structures, lowering cross-linking functionality and ultimately causing network disassembly. Addition of **DAQ** stabilizes MOC structures in the presence of super-stoichiometric Pd²⁺ and subsequently drives MOC formation and gelation. **D** Photographs of sol-gel transitions caused by sequential additions of Pd²⁺ and **DAQ** to **poly(OTf⊂MOC)**. **E** Phase diagram for gelation with 1 equiv. of each guest and varying super-stoichiometric Pd²⁺. The amount of Pd²⁺ that can be tolerated before gel degradation is proportional to guest binding strength. Gelation was assessed by the vial inversion test.

where $f_w$ is the weight-averaged functionality of the MOC network junction; in this case $f_w = 4$, but we note Eq.1 is general for any value of $f_w$. Because $\rho_{L2:PL1}$, $\rho_{L:Pd^{2+}}$, and $f_w$ are fixed according to the initial component stoichiometry and MOC system, it is clear how a shift in ligand self-sorting into MOC structures (i.e., shifting the value of $\phi$ triggered by a small-molecule guest) could cause a shift in the gelation criteria and induce a sol-gel transition. Guided by this analysis, we found that a combination of 1 equiv **PL1** (two ligands per equivalent), 0.25 equiv **L2**, and 1 equiv [Pd(MeCN)₄][OTf]₂ (5 mM Pd²⁺ in DMSO) resulted in a free-flowing liquid even after extended annealing. Upon addition of 1 equiv **DAQ** from a saturated solution in DMSO and further brief annealing (15 min at 50 °C), however, the mixture underwent a sol-gel transition, yielding a free-standing gel with a storage modulus of 2100 ± 50 Pa as measured at 1 rad/s. The subsequent addition of 3

equiv **NO₃** from a saturated DMSO solution caused this material to transform into a free-flowing liquid. Further addition of increasing equivalents of **DAQ** and **NO₃** in an alternating fashion allowed us to achieve a total of 6 sol-gel transitions in a single material (Fig. 6D). A marginal decrease in modulus was observed between each sol-to-gel transition, which we attribute to a combination of slight dilution upon guest solution addition and a reduction of the maximum extent of self-sorting in the presence of the less selective guest, **NO₃**. Nonetheless, we anticipate that the results shown here to trigger a *sol-to-gel* transition for polyMOC gels selectively in the presence of a suitable guest will significantly expand the utility of sol-to-gel transitions as a signal transduction mechanism for sensing applications and provide a general framework for the design of stimuli-responsive materials via guest-biding within tailor-made MOC cavities in general.

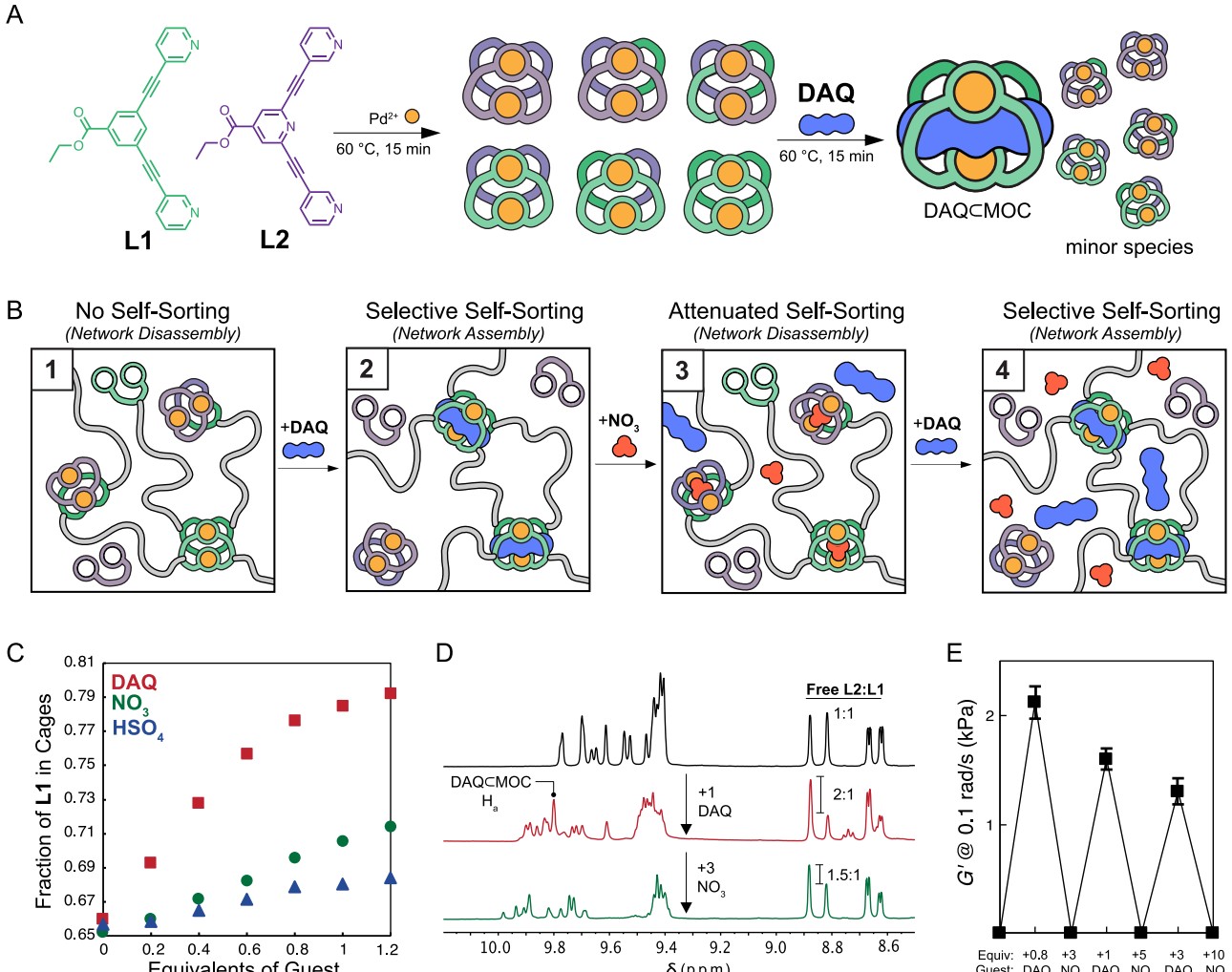

**Fig. 7 | Guest-driven self-sorting in MOCs and polyMOCs containing L1 and L2 moieties. A** Schematic showing the statistical mixture of heteroleptic MOCs initially formed through mixing **L1** and **L2**. When a guest that preferentially binds one of the homoleptic MOC structures is introduced, this homoleptic MOC and subsequent host-guest complex is preferentially enriched relative to other MOC structures. **B** Schematic of self-sorting driven sol-gel transitions within mixtures containing polymer-bound and small molecules ligands (**PL1** and **L2**, respectively) with sub-stoichiometric Pd$^{2+}$. **C** Fraction of **L1** incorporated into MOC structures when **L1** and **L2** are present in equimolar amounts with sub-stoichiometric Pd$^{2+}$ (33.3% excess of bispyridine ligands). The preference to incorporate **L1** into MOC structures increases with strength and equivalents of guest added. **D** $^1$H NMR spectra showing the reversible enrichment of L1 into MOC structures upon inclusion of **DAQ**, followed by depletion when the less selective guest, **NO₃**, is added. **E** Reversible sol-gel transitions achieved through self-sorting behavior. An initial mixture of PL1, L2, and Pd$^{2+}$ form a free-flowing liquid that becomes a gel upon addition of **DAQ**. Error bars represent ±1 standard deviation.

Herein, we have introduced a strategy to incorporate two distinct supramolecular interactions in a nested configuration within a single cross-link of a polymer network based on Pd$_2$L$_4$ MOC junctions. This configuration of metal-ligand coordination and host-guest binding interactions was shown to enable unique functionality inaccessible with networks containing these interaction types in a discrete configuration. The introduction of guest molecules enabled tuning gel relaxation time by over an order of magnitude while maintaining constant stiffness. Further, the introduction and modulation of guest binding strength were shown to enhance the stability of polyMOC gels in the presence of super-stoichiometric Pd$^{2+}$ and competing pyridine-containing ligands, illustrating the synergistic stabilization property of two interactions in parallel within a single network junction. Selective sol-gel transitions gated by guest binding strength could also be realized by simply adjusting the amount of excess Pd$^{2+}$ in the system. Finally, selective guest binding was observed to drive MOC self-sorting, which could be leveraged to enable up to six sequential sol-gel transitions in a single polyMOC gel material composed of a mixture of polymer-bound and small-molecule ligands. A general theoretical description of this phenomenon was presented, which will aid in the extension of these properties to other variations of polyMOC gels. Notably, guest-dependent phenomena all trended proportional to guest binding strength despite their variable ionic nature, suggesting that K$_a$ serves as a general description for tuning such functions and properties in the future. Altogether, these results expand the functionality and stimuli-responsive properties achievable within supramolecular polymer networks through the judicious design of network junction composition and structure. In the future, the nested supramolecular junction concept introduced here may leverage the vast existing body of known MOC host-guest binding properties to accelerate the creation of soft materials with exquisitely selective small-molecule-driven property changes for applications ranging from sensing to biomaterials.

## Methods
### Materials
Solvents were purchased from Millipore Sigma and used as received unless otherwise noted. All aqueous solutions were prepared with

deionized water. Deuterated solvents were purchased from Cambridge Isotopes Laboratories, Inc. Cambridge Isotope Laboratories, Inc. and used as received. 3,5-dibromobenzoic acid, ethyl-3,5,-dibromoisonicotinate, and poly(ethylene glycol) were purchased from Milipore Sigma. 3-Ethynylpyridine was purchased from AmBeed Inc. Tetrakis(acetonitrile)palladium(II) bis(trifluoromethanesulfonate) was purchased from TCI America.

### Nuclear magnetic resonance spectroscopy (NMR)
$^1$H NMR spectra were recorded using either a 400, 500, or 600 MHz Bruker AVANCE spectrometer at room temperature (25 °C). The specific instrument used for each experiment is indicated by the reproduced spectra or listed chemical shifts. All chemical shifts are reported as parts per million (ppm) with splitting patterns designated as follows: s (singlet), d (doublet), t (triplet), q (quadruplet), m (multiplet), and br (broad). Cross polarization magic-angle spinning nuclear magnetic resonance (CP-MAS NMR) spectra were collected using 33.1 μL zirconia spinners. Gels were made as described below, transferred to the spinners, sealed, and then allowed to anneal in the spinner at the indicated time and temperature. Spectrum was collected at a spinning rate of 5000 Hz on a 500 MHz Bruker AVANCE NMR spectrometer.

### Mass spectrometry (MS)
High-resolution mass spectrometry (HRMS) was obtained using QTOF Agilent 6545 mass spectrometer coupled to an Agilent Infinity 1260 LC system running a Jet Stream ESI source. The samples (in acetonitrile) were directly infused using acetonitrile as cosolvent. All measurements were performed at 100 °C dry gas and source temperature and the machine was calibrated in the mass range of 100-3200 Da prior to the measurements. Data analysis was performed with mmass.

### Rheology
Frequency sweep, amplitude sweep, and stress-relaxation experiments were performed using a TA Instruments Discovery HR-2 rheometer with an 8 mm parallel plate geometry unless otherwise noted. Samples were cased using a Teflon mold to a thickness of ~1.5 mm as squares and trimmed into circular pucks with soft plastic tools once loaded on the rheometer. All samples were covered in mineral oil to prevent solvent evaporation and de-swelling. Amplitude sweep experiments were performed on all gels to ensure that the subsequent strain per cent of frequency sweep and stress-relaxation experiments were within the linear viscoelastic regime. All frequency sweep and stress-relaxation experiments were performed at 1% strain. Variable temperature stress-relaxation experiments were performed in 5 °C intervals from 25 °C to 50 °C, and gels were allowed to equilibrate at each temperature for 15 minutes prior to the experiment.

### General MOC Synthesis
L1 or L2 (0.01 mmol, 1 equiv.) was added to a 4 mL vial and dissolved in 300 μL of DMSO-$d_6$. Pd(MeCN)$_4$(OTf)$_2$ (6.27 mg, 0.005 mmol, 1.05 equiv.) was dissolved in 200 μL of DMSO-$d_6$. The palladium solution was added to the polymer ligand solution, and an immediate color change from red-orange to pale yellow was observed, suggesting the formation of palladium-pyridine coordination structures. The solution was characterized immediately by $^1$H NMR. To prepare more concentrated MOC solutions, <300 μL of solvent was used to disperse L1 or L2, to which a palladium stock solution was added. Because the MOCs were generally more soluble in DMSO than the small-molecule ligands, these mixtures would drive ligand solubilization and subsequent MOC formation after stirring for ~15 minutes at room temperature.

### Guest binding studies
The measurements of the guest association constant with MOC were performed by titrating various equivalents of the guest compound into a solution of MOC, maintaining a constant concentration of MOC throughout (5 mM). All studies were done in DMSO, the same solvent used for polyMOC gel fabrication. Both NO$_3$ and HSO$_4$ as guests in MOC and N-MOC displayed rapid exchange with the MOC cavity on the NMR timescale. For these guests, the resonance shifts of one or multiple protons belonging to the MOC were monitored as increasing equivalents of guests were added. These resonance shifts were fit to a 1:1 binding model using a freely available tool BindFit from Supramolecular.org (and P. Thordarson, *Chem. Soc. Rev.*, 2011, *40*, 1305-1323) to calculate the appropriate association constant, K$_a$. Because DAQ exhibited slow exchange with the MOC cavity on the NMR timescale, a modified approach was taken. Upon the addition of DAQ, resonances associated with the guest-bound MOC, DAQ⊂MOC, appeared. The intensity of these resonances relative to those of the un-bound MOC, as determined by integration, could be fit to a 1:1 binding model to calculate the association constant.

### General PolyMOC Gel fabrication procedure
PL1 (55 mg, 0.01 mmol, 1 equiv.) was added to a 4 mL vial and dissolved in 300 μL of DMSO-$d_6$. Pd(MeCN)$_4$(OTf)$_2$ (6.27 mg, 0.0105 mmol, 1.05 equiv.) was dissolved in 200 μL of DMSO-$d_6$. The palladium solution was added to the polymer ligand solution in four portions of 50 μL, with the gel being thoroughly mixed with a spatula and vortex between each addition. After the first two additions, a tacky gel was observed to form. The gel was then mixed for a final time and broken into small pieces before being cast between two Teflon sheets (2 cm×2 cm ×1.5 mm) lined with two pieces of polyethylene for a release liner. The gel casting was annealed at 60 °C for 1 hour to yield a nearly colorless, transparent, homogeneous gel (poly(OTf⊂MOC)).

## Data availability

The data generated or analyzed during this study are available in the article, supplementary information files, or the accompanying source data file. The X-ray crystallography structures reported in this study have been deposited at the Cambridge Crystallographic Data Centre (CCDC), under deposition numbers 2338506, 2338512, 2338513, and 2338517. These data can be obtained free of charge from The Cambridge Crystallographic Data Centre via www.ccdc.cam.ac.uk/data_request/cif. All data are available from the corresponding author upon request. Source data are provided with this paper.

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

## Acknowledgements

D.J.L. acknowledges support from the National Science Foundation Graduate Research Fellowship Program (award #1745320). C.M.B. acknowledges the Natural Sciences and Engineering Research of Canada (NSERC) for a Postdoctoral Fellowship. E.O.B. thanks NWO, The Netherlands Organization for Scientific Research, for funding. This work was partially funded by the Center for the Chemistry of Molecularly Optimized Networks, a National Science Foundation (NSF) Center for Chemical Innovation (CHE-1832256).

## Author contributions

D.J.L., C.M.B., N.J.O., and J.A.J. conceptualized the research. D.J.L., C.M.B., E.O.B., N.J.O., Y.S.A., and J.Z. synthesized compounds. D.J.L., C.M.B., E.O.B., and N.J.O. characterized materials and performed data analysis. I.K. and H.J.K. performed and advised computational studies. D.J.L. and C.M.B. performed crystallographic studies. D.J.L. prepared figures. D.J.L. and J.A.J. wrote the manuscript with input from all co-authors.

## Competing interests

The authors declare no competing interests.
