## [Peer Review File · Nature Communications]

REVIEWER COMMENTS

Reviewer #1 (Remarks to the Author):

In this manuscript, the authors Lundberg et al. report “nested” supramolecular cross-links to influence the properties of cage-based supramolecular organogels. The manuscript was previously submitted to Nat. Chem. And the authors addressed the issues raised by the referees, as discussed in the point-by-point reply. Polymer-linked Pd₂L₄ metal-organic cage gels were employed, which show host-guest binding and metal-ligand coordination within their network junctions. Selective sol-gel transitions governed by guest binding strength and modification of the system by the use of excess Pd²⁺ concentration are presented. It was shown that guest binding induced MOC self-sorting, which could be used to facilitate up to successive sol-gel transitions in a single polyMOC gel material consisting of a blend of ligands and polymer-bound ligands. The manuscript still has issues and is not ready for publication, in its current state.

The authors were confronted with criticism in their last submission, which is still only partially resolved:

Ref 1: “Interactions are leveraged synergistically to control the bulk mechanical properties of gels”, this is correct and novel to some extent, as Nitschke et al. did not study this in JACS 2015. However, host-guest binding within metal-organic cage gels is well (very well) studied and the present manuscript oversells the findings of the authors.

“the influence of guest binding on the (thermo)dynamics of metal–ligand coordination” I do not understand what the authors try to say here, and I fail to see how this was shown in their submission. I’d say the guest binding and metal coordination add up, leading to a change in the bulk property, but I do not see that the guest uptake changes the metal-ligand bond significantly.

“This unique configuration gives rise to a range of functions not before reported in supramolecular networks of any type. “ What functions of the gel are reported here for the first time? What is the unique configuration?

“We note here that all previous polyMOC reports from our group and others, that we are aware of, describe materials constructed solely from metal–ligand coordination / MOC structures. Because these

materials were not designed to intentionally contain multiple supramolecular interactions within a single network junction, and indeed because previous reports (e.g., our own, Nitschke, and Schmidt) do not comment on the influence of hostguest binding on material properties, we believe that it is reasonable to leave them out of Figure 1 (they are not particularly relevant).”

I disagree and feel that the authors insist on comparing apples with oranges. The authors and others have already published a series of polyMOCs, and the interested reader wants to see the novel aspects of this manuscript in comparison to previous publications in the field and not selected contributions from Stang and Craig, which use a different way of constructing polyMOCs. It just doesn't make much sense to me and is confusing to readers. The Johnson group has already published a series of excellent publications on this topic, and this work should be compared to previous relevant works.

Speaking of apples and oranges, the other very relevant but overlooked part of the lengthy host-guest discussion, together with the issues raised regarding the titrations: HSO₄ and NO₃ are partial counter-anion exchanges of the fourfold positively charged Pd-cage. This is compared to the guest uptake of neutral DAQ. The mechanism is inherently different, and I would suggest the authors compare the DAQ uptake to another aromatic guest, as this does make only so much sense now in the current manuscript and also explains why the titrations are done the way they are done.

“We wish to note that while indeed the Pd₂L₄ motif has been established, the acetylene-spaced Pd₂L₄ motif has not previously been reported in the context of polyMOCs” This is not a strong selling point for a submission to NatCommun.

Stress relaxation studies and everything onwards from there are addressed sufficiently, except issues with the SI and general writing of the manuscript are addressed, which is also a point raised by referee 2, reviewer 2-3, the subscript and superscript, as well as other formatting that is still all over the place in the manuscript and SI.

The manuscript is improved in some parts of the introduction, however, errors persist, which is frustrating. The pages 4–8 in the submission are lengthy before the modulation starts.

Unify the subscript on page 1! HSO₄ subscript, NO₃, how many times do two referees ask for something like this? Also, check the whole SI document carefully, please.

Justify all paragraphs, not just some paragraphs randomly!

Throughout the whole SI.

Throughout the SI, please use subscript on DMSO-d₆ and unify the writing of it

The formatting of each NMR is different. Fig S8. Black and white, no title, Fig S10. Coloured, no name, no integrals picked! Please unify the formatting of all DOSYs, too.

SI: p3 it is still an avance nmr spectrometer

Fig S1. Please label and assign all peaks, use the scheme used in the manuscript, not 1, 2, 3, also spaces. Show the molecule, so that readers understand the assignment. Chemical shifts, not sifts.

Fig S2. Same.

Fig S3. Same.

Fig S4. NO₃ subscript, assign protons and label in a meaningful way.

Fig. S10. Two z, but no w! H₂O subscript....

Fig S15. Quality of the DOSY is very poor. Crosspeaks for the ethyl group are missing. Fomattng of Ha*)

Fig S16. Please either use integrals or multipllett analysis for ALL NMRs consistently.

P 20, Ethyl-3,5-dibormobenzoate significant figures all over the place

¹H NMR accounts for 12 protons, 16 are expected!

Fig S18. See above NMR text

P 26, Polymer Ligand (PL) formatting, ¹H 7.92 counted twice?

Fig S23. F is labelled twice, should be e once

Reviewer #2 (Remarks to the Author):

The authors constructed a class of supramolecular gels which contained both metal -ligand coordination and host-guest interaction, calling “nested” supramolecular crosslinks. A series of characterization analysis including NMR titrations, solid-state NMR and X-ray analysis were studied the influences on the host-guest interaction in supramolecular gels and reversible sol-gel transitions could be realized through the host-guest interaction. I think this work expand the functionality and stimuli-responsive properties achievable within supramolecular polymer networks through the judicious design of network junction chemistry and structure. However, major revision is necessary before publication. The issues and suggestions are listed as follows:

1. I agree with the suggestions of Reviewer 2 about the characters the performance of the gels in the process of "PolyMOC Gels Stabilized by Guests in the Presence of Superstoichiometric Pd²⁺". And the authors gave a detailed reply to it in Figure S31, but there are still many problems. In Figure S31a, is it the assembly with varying equivalents of Pd²⁺ (1 to 5 equiv.) ? The assembly with 5 or 0.5 equivalents is unreasonable. In Figure S31b, what is the G3 ? In addition, how does the stress relaxation data of polyMOC gels compare with the polyMOC gels containing 3 equiv. Pd²⁺ with and without 1 equiv. DAQ? The authors should provide an explanation and add a statement in the manuscript.

2. To better illustrate the self-sorting response to the gels, the stress relaxation studies about the gels of selective self-sorting should be added.

3. The ¹H CP-MAS NMR spectrum of the gel of Poly(DAQ⊂MOC) in the presence of excessive Pd²⁺ should be added.

4. In Figure S15A, ¹H DOSY NMR of DAQ⊂MOC should be retested, the signal peaks of the guest molecule are not in line with the MOC at all. In addition, the correlation between the MOC and DAQ in Figure S15B should be clearly labeled.

5. I think literature about the assemblies based on supramolecular gels should be cited (Sci. China: Chem. 2021, 64, 1177-1183; Chem. Eur. J. 2023, 29, e2023030; Sci. China: Chem. 2023, 10.1007/s11426-023-1872-5)

6. The authors modified a small part for the problem of "NO₃ and HSO₄ sometimes subscript, sometimes not, should be unified, also in Figures and SI (captions, headlines). In general, careful formatting of the SI, bold brackets, Pd²⁺ should always be superscript, etc." However, the same problem still exists in the manuscript and the supporting information.

Reviewer #3 (Remarks to the Author):

I would now be happy to recommend publication in Nature Communications following the authors' changes.

Reviewer #1 (Remarks to the Author):

In this manuscript, the authors Lundberg et al. report “nested” supramolecular cross-links to influence the properties of cage-based supramolecular organogels. The manuscript was previously submitted to Nat. Chem. And the authors addressed the issues raised by the referees, as discussed in the point-by-point reply. Polymer-linked Pd₂L₄ metal-organic cage gels were employed, which show host-guest binding and metal-ligand coordination within their network junctions. Selective sol-gel transitions governed by guest binding strength and modification of the system by the use of excess Pd²⁺ concentration are presented. It was shown that guest binding induced MOC self-sorting, which could be used to facilitate up to successive sol-gel transitions in a single polyMOC gel material consisting of a blend of ligands and polymer-bound ligands. The manuscript still has issues and is not ready for publication, in its current state.

The authors were confronted with criticism in their last submission, which is still only partially resolved: Ref 1: “Interactions are leveraged synergistically to control the bulk mechanical properties of gels”, this is correct and novel to some extent, as Nitschke et al. did not study this in JACS 2015. However, host-guest binding within metal-organic cage gels is well (very well) studied and the present manuscript oversells the findings of the authors.

We appreciate the reviewer’s comment. We humbly suggest that they clarify and/or provide references to support their claims. To our knowledge, there are very few examples of host-guest binding in polyMOCs (at most 4 papers to our knowledge), and none that show how these binding interactions can be leveraged to achieve the properties demonstrated in this manuscript.

“the influence of guest binding on the (thermo)dynamics of metal–ligand coordination” I do not understand what the authors try to say here, and I fail to see how this was shown in their submission. I’d say the guest binding and metal coordination add up, leading to a change in the bulk property, but I do not see that the guest uptake changes the metal-ligand bond significantly.

We humbly suggest that the data presented throughout our paper support this statement. For example, it is known that network relaxation is connected to the kinetics of metal–ligand coordination. Here, introducing a guest changes the network relaxation rate. Thus, the binding of the guest to the host changes the rate of metal–ligand bond exchange, changing network relaxation. This affect is a function of the binding constant (thermodynamics). We also give numerous examples of how guest binding changes the host equilibrium constant. For example, both in the presence of super stoichiometric Pd²⁺ and under mixed-ligand conditions (to drive self-sorting), guest binding changes the preferred type of metal-ligand bond/coordination (shifts equilibrium; thermodynamics).

We note that this difference is key to the concept introduced in this paper. The effects are linked in *parallel*, such that host-guest binding changes the rate of metal–ligand exchange. In conventional systems with 2 interactions *in series*, changing one does not necessarily change thermodynamics of the other. Similarly, for systems in series, breaking one interaction breaks the network. In our system, removing the guest (i.e., removing the host-guest interaction) does not break the network.

“This unique configuration gives rise to a range of functions not before reported in supramolecular networks of any type. “ What functions of the gel are reported here for the first time? What is the unique configuration?

The unique configuration is the two interactions in *parallel* within a *single* cross-link of the gel (as opposed to in *series*, which is common). Each demonstrated function in the paper is reported here for the first time, as stated in our paper. As one example, being able to form gels far from stoichiometry is impossible for other

networks; here, it is possible to use as many as 5 equivalents and still form gels due to the altered thermodynamics that occur upon guest binding. Several other examples are given throughout the manuscript.

To attempt to further clarify the key differences for this reviewer, we have drawn the simplified schematic shown below. On the top, we show a *parallel* arrangement of two supramolecular interactions within a single network cross-link. On the bottom, we show the common *series* arrangement.

“Nested” Supramolecular Crosslinks (this work)

Properties and responsiveness determined by the **combination** of both interactions

“Discrete” Supramolecular Crosslinks (previous work)

Properties and responsiveness determined by interactions **independent of each other** (i.e., dynamics and degradation determined by *weakest link*)

“We note here that all previous polyMOC reports from our group and others, that we are aware of, describe materials constructed solely from metal–ligand coordination / MOC structures. Because these materials were not designed to intentionally contain multiple supramolecular interactions within a single network junction, and indeed because previous reports (e.g., our own, Nitschke, and Schmidt) do not comment on the influence of hostguest binding on material properties, we believe that it is reasonable to leave them out of Figure 1 (they are not particularly relevant).”

I disagree and feel that the authors insist on comparing apples with oranges. The authors and others have already published a series of polyMOCs, and the interested reader wants to see the novel aspects of this manuscript in comparison to previous publications in the field and not selected contributions from Stang and Craig, which use a different way of constructing polyMOCs. It just doesn't make much sense to me and is confusing to readers. The Johnson group has already published a series of excellent publications on this topic, and this work should be compared to previous relevant works.

We reiterate that our group's previous examples on polyMOC gels have only investigated materials wherein metal–ligand interactions are used to form the network. In the present work, we are not focused on polyMOCs in general, but on the concept of materials with *two distinct supramolecular interactions* that exist *in parallel* within single cross-links (see simplified schematic above). We do not compare to other polyMOCs because other polyMOCs do not have this feature.

We compare to Stang and Craig because their works involve 2 different supramolecular interactions, making them the most relevant comparisons; however, they use two different supramolecular interactions *in series* or in separate cross-links to construct polymer networks, which is fundamentally different. Here, we clearly show the unique properties that can be achieved by having two supramolecular interactions *in parallel* that would not be possible if they are *in series*.

This work describes a concept—*materials constructed with nested, parallel interactions*—that is not limited to polyMOCs. We only use polyMOCs here to demonstrate that concept, but it could be envisioned in other classes of materials. Thus, it does not make sense, in our humble opinion, to compare to other polyMOCs (or other materials in general) that do not rely on two interactions for their assembly.

Speaking of apples and oranges, the other very relevant but overlooked part of the lengthy host-guest discussion, together with the issues raised regarding the titrations: HSO₄ and NO₃ are partial counter-anion exchanges of the fourfold positively charged Pd-cage. This is compared to the guest uptake of neutral DAQ. The mechanism is inherently different, and I would suggest the authors compare the DAQ uptake to another aromatic guest, as this does make only so much sense now in the current manuscript and also explains why the titrations are done the way they are done.

Our intent is not to distinguish between effects related to the precise mechanism of guest uptake (i.e., differences between ionic guests and neutral guests) on host-guest modulation of polyMOC properties; instead, our focus on binding *strength* we believe provides more general insights into this new mechanism of material property control and modulation. Further, despite potential differences between binding of neutral and ionic guests, trends of relaxation time (Figure 5A), stability in the presence of excess Pd²⁺ (Figure 6E), and self-sorting efficiency (Figure 7C) all trend with binding *strength*.

“We wish to note that while indeed the Pd₂L₄ motif has been established, the acetylene-spaced Pd₂L₄ motif has not previously been reported in the context of polyMOCs” This is not a strong selling point for a submission to NatCommun.

We appreciate the reviewer’s opinion, though we do not believe we have claimed that this motif is a “strong selling point” for submission. The selling point is that this work describes the design of networks with “*nested*” or *parallel* supramolecular interactions (which are enabled by this MOC motif) and how such a design can enable material functions that are not possible for traditional supramolecular networks.

Stress relaxation studies and everything onwards from there are addressed sufficiently, except issues with the SI and general writing of the manuscript are addressed, which is also a point raised by referee 2, reviewer 2-3, the subscript and superscript, as well as other formatting that is still all over the place in the manuscript and SI.

Consistent formatting and subscript/superscripting has been added throughout the revised manuscript and SI as requested by the reviewer (e.g., DMSO-d₆, NO₃, HSO₄). We thank the reviewer for their very thorough review of the SI.

The manuscript is improved in some parts of the introduction, however, errors persist, which is frustrating. The pages 4–8 in the submission are lengthy before the modulation starts.

We believe thorough characterization of the host-guest binding in the small-molecule MOCs is justified to provide a strong foundation for the study of how guest binding ultimately influences polyMOC bulk properties, particularly as guest binding under the conditions explored here (e.g., highly donating solvent) have not been explored in depth previously. We do not believe we should jump straight to the final highlights of what is possible with nested interactions without first describing the materials and characterizing them carefully.

Unify the subscript on page 1! HSO₄ subscript, NO₃, how many times do two referees ask for something like this? Also, check the whole SI document carefully, please.

The subscripts have been unified for “HSO₄” and “NO₃” throughout the revised main text and SI.

Justify all paragraphs, not just some paragraphs randomly! Throughout the whole SI.

All text and figure captions have been consistently justified throughout the revised SI.

Throughout the SI, please use subscript on DMSO-d₆ and unify the writing of it

Subscripts on "DMSO-d₆" have been changed throughout the revised SI for consistency.

The formatting of each NMR is different. Fig S8. Black and white, no title, Fig S10. Coloured, no name, no integrals picked! Please unify the formatting of all DOSYs, too.

The formatting of NMRs throughout the SI has been changed for consistency.

SI: p3 it is still an avance nmr spectrometer

This has been updated in the Materials and Methods section of the revised SI.

Fig S1. Please label and assign all peaks, use the scheme used in the manuscript, not 1, 2, 3, also spaces. Show the molecule, so that readers understand the assignment. Chemical shifts, not sifts.

Fig S2. Same.

Fig S3. Same.

Fig S4. NO₃ subscript, assign protons and label in a meaningful way.

Chemical structures of host-guest complexes and their assignments relating to chemical shifts in titration experiments have been added to Figures S1-S4 in the revised SI.

Fig. S10. Two z, but no w! H₂O subscript....

The spectra labels have been updated in the revised SI.

Fig S15. Quality of the DOSY is very poor. Crosspeaks for the ethyl group are missing. Fomattng of Ha*)

A new spectrum has been acquired with higher quality and included in the revised SI.

Fig S16. Please either use integrals or multipllett analysis for ALL NMRs consistently.

All NMR analysis has been conducted in a consistent form throughout the revised SI.

P 20, Ethyl-3,5-dibormobenzoate significant figures all over the place
1H NMR accounts for 12 protons, 16 are expected!

The experimental write-up and NMR labeling have been updated in the revised SI.

Fig S18. See above NMR text

The new consistnt formatting of NMR spectra has removed such text throughout the revised SI.

P 26, Polymer Ligand (PL) formatting, 1H 7.92 counted twice?

This was an error, peaks at 7.92 and 7.94 ppm are observed and are correctly reported in the revised SI.

Fig S23. F is labelled twice, should be e once

The labeling of this spectra has been updated in the revised SI.

Reviewer #2 (Remarks to the Author):

The authors constructed a class of supramolecular gels which contained both metal-ligand coordination and host-guest interaction, calling “nested” supramolecular crosslinks. A series of characterization analysis including NMR titrations, solid-state NMR and X-ray analysis were studied the influences on the host-guest interaction in supramolecular gels and reversible sol-gel transitions could be realized through the host-guest interaction. I think this work expand the functionality and stimuli-responsive properties achievable within supramolecular polymer networks through the judicious design of network junction chemistry and structure. However, major revision is necessary before publication. The issues and suggestions are listed as follows:

1. I agree with the suggestions of Reviewer 2 about the characters the performance of the gels in the process of “PolyMOC Gels Stabilized by Guests in the Presence of Superstoichiometric Pd²⁺”. And the authors gave a detailed reply to it in Figure S31, but there are still many problems. In Figure S31a, is it the assembly with varying equivalents of Pd²⁺ (1 to 5 equiv.) ? The assembly with 5 or 0.5 equivalents is unreasonable. In Figure S31b, what is the G3? In addition, how does the stress relaxation data of polyMOC gels compare with the polyMOC gels containing 3 equiv. Pd²⁺ with and without 1 equiv. DAQ? The authors should provide an explanation and add a statement in the manuscript.

The characterization data presented in this figure is for MOC assembly (panel A), and polyMOC assembly (panel B, judged by storage modulus measurement at 1 rad/s) with between 1 and 5 equivalents of Pd²⁺. A brief description has been added above the figure to make this clearer. We note that no assembly with 0.5 equiv Pd²⁺ is shown in our work; however, we do present assembly with 0.5 equiv DAQ as a single data point (grey dot) in Figure S31A. Further, we show that in some cases assembly with 5 equiv Pd²⁺ is *not* unreasonable due to the stabilizing effect of guest binding (Figure S31B and Figure 6E). We note that this unexpected finding is a key point of this paper as it shows the unprecedented results that can be achieved with this nested junction design.

We regret the error in labeling DAQ as G3 in Figure S31B and have updated this in the revised manuscript. Regarding the stress relaxation data, we find that gels containing 3 equiv. Pd²⁺ display a shorter relaxation time than on-stoichiometry assembly, likely due to their lower modulus and higher fraction of unbound ligands, resulting in lower cross-linker functionality networks, which are known to display shorter relaxation times than higher functionality networks. Addition of guests under these super stoichiometric Pd²⁺ conditions leads to an increase in relaxation time and a concomitant increase in modulus (as shown in Figure S31B); however, MOC assembly is not *completely* recovered and thus the relaxation time is still moderately shorter than on-stoichiometry polyMOC containing DAQ (which is expected). This discussion has been added to the revised SI and the main text has been updated to mention this study (in addition to the newly acquired CP-MAS NMR data).

2. To better illustrate the self-sorting response to the gels, the stress relaxation studies about the gels of selective self-sorting should be added.

Stress relaxation studies of the gels from the self-sorting experiments have been added as Figure S35. In this case, direct comparison to other polyMOC systems in this work is complicated by the fact that self-sorting systems contained an appreciable fraction of L2 and were assembled with sub-stoichiometric Pd²⁺, thus they displayed markedly lower moduli and relaxation times, even in the presence of DAQ. This figure and brief discussion have been added to the “Guest-Triggered Sol-Gel Transitions” section of the revised SI.

Figure S35. Representative stress relaxation studies of the polyMOC gels from the self-sorting studies shown in Figure 7E.

3. The ^1H CP-MAS NMR spectrum of the gel of Poly(DAQ-MOC) in the presence of excessive Pd^{2+} should be added.

These spectra have been added to the revised supporting information (Figures S32-S34). Resonance shifts indicative of guest binding are observed when DAQ is added under super stoichiometric Pd^{2+} conditions. Additional discussion has been added for this data in the “*Characterization of Gels Fabricated with Excess Pd^{2+}* ” section of the revised SI highlighting the indicators of guest binding and comparing these data to analogous small-molecule MOC studies presented in Figure 6 of the main text.

4. In Figure S15A, ^1H DOSY NMR of DAQ-MOC should be retested, the signal peaks of the guest molecule are not in line with the MOC at all. In addition, the correlation between the MOC and DAQ in Figure S15B should be clearly labeled.

A new DOSY spectra has been acquired with higher quality and the correlation between MOC and DAQ in Figure S15B has been clearly labeled in the revised SI.

5. I think literature about the assemblies based on supramolecular gels should be cited (Sci. China: Chem. 2021, 64, 1177-1183; Chem. Eur. J. 2023, 29, e2023030; Sci. China: Chem. 2023, 10.1007/s11426-023-1872-5)

These references have been added to the main text when discussing such materials (Ref. #10 to 12).

6. The authors modified a small part for the problem of “ NO_3 and HSO_4 sometimes subscript, sometimes not, should be unified, also in Figures and SI (captions, headlines). In general, careful formatting of the SI, bold brackets, Pd^{2+} should always be superscript, etc.” However, the same problem still exists in the manuscript and the supporting information.

We regret the persistence of errors in our manuscript and have thoroughly reviewed the revised documents to ensure consistent formatting, subscripting, and captions (e.g., subscripting DMSO-d_6 , NO_3 , HSO_4 , consistent formatting of NMR spectra, and justification of paragraphs in the SI).

Reviewer #3 (Remarks to the Author):

I would now be happy to recommend publication in Nature Communications following the authors' changes.

We thank the reviewer for considering our manuscript.

REVIEWER COMMENTS

Reviewer #1 (Remarks to the Author):

The authors present a partially revised manuscript. Addressing some of the comments made by the referees. The authors have an interesting case here, and many of the things they write in their referee response are correct, yet for some reason I do not understand why they are not reflected in the manuscript. The abstract and the introduction are confusing and partially incorrect, mixing supramolecular polymers with non-covalent interactions and metal-ligand coordination without defining such important key terms early on.

The discussion about supramolecular interactions in series or in parallel reads still so outlandish, as if host-guest interactions would be a single interaction and not an interplay of different weak interactions, such as hydrogen bonding between guest and cage, and so on. The authors claim that other supramolecular polymers do not exhibit such features, which is not correct.

The authors are indeed the first to study the guest binding effect on some of the macroscopic properties of the polyMOCs. However, the authors compare neutral guest binding to partial anion exchange, which does not make much sense and has no benefit to the reader. The same holds true for Figure 1, which misleads the general reader away from previous works of the authors and other groups. The guest binding discussion up until page 9 is very well established, and there is no need to discuss all of this lengthy in the main manuscript. The other parts are actually interesting and will become worth publishing if the manuscript is written accordingly.

Reviewer #2 (Remarks to the Author):

Comments to the Author

The author has given a reasonable explanation to the questions raised by the reviewer. I agree to publish this article on Nature Communications.

Reviewer #1 Comments

The authors present a partially revised manuscript. Addressing some of the comments made by the referees. The authors have an interesting case here, and many of the things they write in their referee response are correct, yet for some reason I do not understand why they are not reflected in the manuscript. The abstract and the introduction are confusing and partially incorrect, mixing supramolecular polymers with non-covalent interactions and metal-ligand coordination without defining such important key terms early on.

We thank the reviewer for their comments. We have updated our abstract and introduction to further clarify the purpose of this work.

The discussion about supramolecular interactions in series or in parallel reads still so outlandish, as if host-guest interactions would be a single interaction and not an interplay of different weak interactions, such as hydrogen bonding between guest and cage, and so on. The authors claim that other supramolecular polymers do not exhibit such features, which is not correct.

The reviewer is correct that host-guest (and indeed many non-covalent) interactions involve an interplay of many weak interactions. We do not claim otherwise. We note that because such ensemble interactions are defined by, *e.g.*, the host/guest pair or the metal–ligand bond, and cannot be modified independently, they are commonly viewed as an ensemble or as a single interaction type (*e.g.*, a “host-guest” interaction is referred to as a single type of interaction though it involves many weak non-covalent forces). We had perhaps erroneously assumed that this distinction did not need definition; however, we have revised our introduction to make it clearer to the general reader. The key point of our paper is that we use multiple *interaction types* (host-guest *and* metal–ligand coordination, each of which may involve an interplay of multiple weak interactions) *in parallel* (rather than series) to achieve new material functions.

The authors are indeed the first to study the guest binding effect on some of the macroscopic properties of the polyMOCs. However, the authors compare neutral guest binding to partial anion exchange, which does not make much sense and has no benefit to the reader.

Our observations that guest-triggered material property changes trend with binding strength (as quantified by association constants, K_a) strongly suggest that despite potential differences in the *nature* of the binding interaction (*e.g.*, ionic or not), overall binding strength serves as a general descriptor for the effects on material properties (demonstrating generality beyond a single interaction class). We have updated the manuscript to mention the potential different contributions to guest binding (*e.g.*, electrostatic for ionic guests) during our discussion of guest binding and crystal structures but emphasize that in all cases the guests have strong and selective interactions with the MOC cavity.

The same holds true for Figure 1, which misleads the general reader away from previous works of the authors and other groups. The guest binding discussion up until page 9 is very well established, and there is no need to discuss all of this lengthily in the main manuscript. The other parts are actually interesting and will become worth publishing if the manuscript is written accordingly.

Our introduction mentions previous results regarding polyMOCs from our group, and it has been updated further to highlight how supramolecular structures as cross-links (*e.g.*, MOCs or metallacycles) based upon a single class of non-covalent interaction (in the case of polyMOCs, metal–ligand coordination) display material and stimuli-responsive properties defined by the underlying non-covalent interaction. The following sentences have been added to the introduction of our manuscript:

*“In some cases, non-covalent bonds form supramolecular structures (*e.g.*, metallacycles from metal-ligand coordination bonds or extended pi-pi stacking clusters) which themselves can serve as cross-links. While such structures can introduce advanced functionality, their mechanical properties and stimuli-responsiveness are generally defined by the underlying non-covalent bonds.”*

Again, we feel that detailed discussion regarding guest binding in our system is warranted because of the unique properties of our system compared to previous studies (*e.g.*, highly donating solvent, binding counter-ions). Additionally, detailed and thorough comparisons of guest binding in polyMOCs with their corresponding MOCs

provides a strong foundation for all claims and experiments later in the manuscript relating to guest-triggered property changes.